# Lower Glucose Effectiveness Is Associated with Subclinical Reactive Hypoglycemia, Snacking Habits, and Obesity

**DOI:** 10.3390/metabo13020238

**Published:** 2023-02-06

**Authors:** Ichiro Kishimoto, Akio Ohashi

**Affiliations:** 1Department of Endocrinology and Diabetes, Toyooka Public Hospital, 1094 Tobera, Toyooka 668-8501, Japan; 2NEC Corporation, Environment and Total Quality Management Division, 5-7-1 Minato-ku, Shiba, Tokyo 108-0014, Japan

**Keywords:** subclinical reactive hypoglycemia, glucose effectiveness, obesity, overweight, continuous glucose monitoring

## Abstract

The effects of glucose effectiveness, the insulin-independent mechanism of glucose disposal, on hypoglycemia have not yet been fully investigated. Herein, in 50 males without a diagnosis of diabetes mellitus (median age 54 years, body mass index (BMI) ≥ 25), the index of glucose effectiveness (SgIo) was determined by a 75 g oral glucose tolerance test (OGTT), and continuous glucose monitoring (CGM) was performed for 6 days. The minimal glucose levels and the percentages of time below 70 mg/dL (3.9 mmol/L) (TBR70) during CGM were significantly associated with the SgIo tertile category in a biphasic manner. When TBR70 within 24 h after OGTT ≥ 0.6% was defined as subclinical reactive hypoglycemia (SRH), odds ratios of having SRH in SgIo tertile 1 (lowest) and tertile 3 (highest) compared to SgIo tertile 2 (middle) were both 11.7 (*p* = 0.007), while the odds ratios of the highest post-load insulin quartile were 22.9 (*p* = 0.001) and 1.07 (*p* = 0.742), respectively. The chances of having self-reported snacking habits, obesity (BMI ≥ 30), and impaired glucose tolerance were significantly higher in participants in SgIo tertile 1 compared to those in SgIo tertile 2, with odds ratios of 10.7 (*p* = 0.005), 11.2 (*p* = 0.02), and 13.8 (*p* = 0.002), respectively. However, there was no significant difference between SgIo tertile categories 2 and 3. In conclusion, SgIo is associated with SRH in a biphasic manner. In people with lower glucose effectiveness, the SRH-induced increase in appetite may create a vicious cycle that leads to obesity.

## 1. Introduction

Hypoglycemia is characterized by a decrease in plasma glucose concentration to a level that induces neurogenic or neuroglycopenic symptoms. Although the most common cause is diabetes medication, including sulfonylureas and exogenous insulin, symptomatic hypoglycemia can occur in people who do not have diabetes mellitus [1]. Some of these individuals exhibit post-meal hypoglycemia with autonomic nervous symptoms, which is called reactive hypoglycemia [2]. Reactive hypoglycemia may be due to dysregulated insulin secretion or increased insulin sensitivity, which often has no organic lesions. It is reported that obese people have a higher rate of symptomatic reactive hypoglycemia than non-obese people after a 75 g oral glucose load [3].

Using the hyperinsulinemic glucose clamp, activation of autonomic symptoms such as palpitations, tremors, and sweating develops at a plasma glucose concentration of approximately 58 mg/dL (3.2 mmol/L) [4]. Recently, we have examined diurnal glycemic patterns using continuous glucose monitoring (CGM) in obese or overweight subjects without diabetes mellitus, and we reported that more than half of the participants exhibited low glucose levels (<70 mg/dL (3.9 mmol/L)) after a 75 g glucose load, which did not provoke typical adrenergic symptoms of hypoglycemia [5]. Nonetheless, the snacking frequency of the participants with subclinical (asymptomatic except hunger) reactive hypoglycemia (SRH) was significantly higher than that of the participants without SRH [5], indicating that glucose levels that are lower than the lower limit of the normal range (70 mg/dL (3.9 mmol/L)), but higher than the hypoglycemic threshold for activation of autonomic symptoms (approximately 58 mg/dL (3.2 mmol/L)), still induce appetite and lead to an increased frequency of snacking in daily life. Therefore, SRH may play a key role in the establishment and/or maintenance of obesity, and studying the mechanism of SRH is of clinical significance.

Although the specific causes have not been fully determined, too much insulin in the bloodstream at the incorrect time is thought to be involved in symptomatic reactive hypoglycemia. However, in our previous study, the associations between SRH and insulin-related indices, such as homeostatic model assessment (HOMA)-β, HOMA-R, Matsuda index, disposition index, or insulinogenic index, were minor [5], suggesting that a mechanism other than delayed hyperinsulinemia might be involved in SRH.

Glucose effectiveness is an insulin-independent mechanism of glucose disposal from blood circulation, which is a major contributing factor for intravenous glucose tolerance [6] and has been considered as an important mechanism for the maintenance of normoglycemia [7]. It is known that glucose effectiveness is reduced in subjects with type 2 diabetes [7] or obesity [8,9]. We have previously reported that during CGM, a substantial proportion of obese/overweight men without diabetes mellitus exhibited elevated post-meal glucose levels above the recommended target of diabetes treatment [10,11]. We also observed that lower SgIo, a 75 g oral glucose tolerance test (OGTT)-derived index for glucose effectiveness, was associated with hyperglycemia in the population independently of blood insulin levels [12]. However, the role of glucose effectiveness on SRH has not yet been studied. Therefore, in the present study, we investigated the relationship between SgIo and SRH.

## 2. Methods

Study protocol: The study protocol has been described elsewhere [11]. In brief, 50 male participants (body mass index (BMI) ≥ 25 kg/m^2^, age 50–65 years) were recruited (mid-life men with obesity or overweight were studied, since this population is at high risk of developing future diabetes mellitus). After overnight fasting (≥12 h), blood samples for fasting plasma glucose (FPG), serum glycated hemoglobin (HbA1c), and plasma 1,5-anhydroglucitol (AG) were drawn from the cubital vein, and a 2 h 75 g OGTT was performed. After the OGTT, self-monitoring of a seven-point blood glucose (BGM) profile (preprandial, 1~2 h postprandial, and pre-bedtime) was performed using a glucometer (Glutest Neo Alpha; Sanwa Kagaku Kenkyusho Co., Osaka, Japan) every day during the study period (6 days). The participants were also instructed to wear CGM devices (iPro™2 Professional CGM, Medtronic, MN, USA) during the study. The CGM sensor is designed to collect the glucose information in the interstitial fluid and send a reading, which is retrieved by the transmitter. The sensor was calibrated at least four times throughout the day, according to the manufacturer’s specifications. On the 1st day, the participants were asked to answer as to whether they ate snacks regularly with a self-reported questionnaire. During the study, eating, drinking, and exercise were at the discretion of the participants.

Glucose effectiveness-related index: SgIo, an index for glucose effectiveness, was calculated from 75 g OGTT data using Nagasaka’s equation [13]. In brief, SgIo (mg/dL/min) = [(PPG (post-loading plasma glucose) without insulin and glucose effectiveness) − (PPG without insulin/with glucose effectiveness) × (adjustment factor)]/120

where:

(PPG without insulin and glucose effectiveness) = FPG (mg/dL) + (0.75 × 75,000)/(0.19 × body weight in kg × 10).

(PPG without insulin/with glucose effectiveness) = 152, 213, and 342 mg/dL for normal glucose tolerance (NGT), impaired glucose tolerance (IGT), and diabetes mellitus (DM), respectively.

(adjustment factor) = 2hPG(plasma glucose at 2h post-OGTT)/2hPGE (expected 2hPG), where

2hPGE = 124.1 × 24.4[log_10_ DIo (disposition index determined by OGTT data)] for subjects with NGT;

2hPGE = 160.8 × 44.8[log_10_ DIo] for subjects with IGT;

2hPGE = 211.6 × 112.1[log_10_ DIo] for subjects with DM

Insulin-related indices: Using plasma glucose concentration (PG) in mg/dL and immunoreactive serum insulin concentration (IRI) in μU/mL under fasting conditions or during the OGTT, indices of insulin secretion or insulin resistance were calculated indirectly using the following formulas.

Homeostatic model assessment (HOMA)-β = OGTT IRI at 0 min × 360/(OGTT PG at 0 min − 63).

HOMA-R = (OGTT PG at 0 min × OGTT IRI at 0 min)/405

insulinogenic index = (OGTT IRI at 30 min − OGTT IRI at 0 min)/(OGTT PG at 30 min − OGTT PG at 0 min)

Matsuda index = 10,000/SQRT((OGTT PG at 0 min × OGTT at IRI 0 min) × ((OGTT PG at 0 min + OGTT PG at 30 min × 2 + OGTT PG at 60 min × 3 + OGTT PG at 120 min × 2)/8 × (OGTT at IRI 0 min + OGTT IRI at 30 min × 2 + OGTT IRI at 60 min × 3 + OGTT IRI at 120 min × 2)/8)), respectively.

Data analysis for CGM: As indicators of low glucose levels, the minimal CGM sensor glucose level (CGM min), as well as the percentage of time below the range when glucose levels were <70 mg/dL (3.9 mmol/L) (TBR70), were determined using all CGM glucose data obtained during the study. The participants with CGM readings ≥720 (60 h) were chosen for the analysis including CGM data (*n* = 43).

Definition of SRH: When the logistic regression model was constructed to examine the relationships between TBR70 values within 24 h after an oral glucose load, TBR70 (24 h), and the snacking habits category, a 1% increase in TBR70 (24 h) was associated with an 8% increase in the risk of having snacking habits. The receiver operating characteristic (ROC) curve analysis of TBR70 (24 h) for detecting snacking habits revealed an area under the curve (AUC) of 0.68 and a cutoff of 0.6% (sensitivity 69%, selectivity 67%). The present study, therefore, defined TBR70 (24 h) ≥0.6% as SRH. Daily logs of subjective symptoms during the study revealed no specific hypoglycemic symptoms except hunger.

Statistics: Baseline characteristics are presented in this paper as the median (interquartile range; IQR) according to the SgIo tertile. Three group differences were statistically analyzed using the Kruskal–Wallis test. If significant differences were found among groups, pairwise comparisons were tested via the Steel–Dwass multiple comparisons test. For categorical data, Pearson’s chi-squared test was used to determine whether there was an association between the proportions of the participants and the SgIo tertile category. For the association between the proportions of the participants with SRH and the SgIo tertile categories, the Cochran–Armitage trend *p* was calculated in each OGTT IRI 120 tertile. In addition, when the number of each category was small, a one-sided Fisher’s exact probability test with post hoc Bonferroni corrections for multiple comparisons was applied to investigate whether there were any differences between SgIo tertile 1 (the lowest category) or SgIo tertile 3 (the highest category) and SgIo tertile 2 (the middle category). The level of significance was set to 5%, and *p*-values of less than 0.05 were considered statistically significant. When the highest or lowest SgIo category was compared to the middle SgIo category with Fisher’s exact probability test, *p*-values of less than 0.025 were considered statistically significant.

Ethics: The study was approved by the institutional review board of Toyooka Public Hospital (#146; 3 October 2017) and the Japan Conference of Clinical Research review board (JCCR#3-132; 21 October 2016). Written informed consent was taken from all participants before study enrollment. This study was performed in accordance with the principles established by the Helsinki Declaration.

## 3. Results

### 3.1. Characteristics of the Participants According to the SgIo Tertile Category

Baseline characteristics according to the SgIo tertile category are shown in Table 1. When the participants were divided by SgIo tertile with cut points of 2.388 and 2.769 mg/dL/min, BMI and HbA1c levels were significantly associated with the SgIo category. In addition, glucose and insulin levels at 2 h after a 75 g glucose load increased with decreasing SgIo tertiles. Furthermore, the SgIo tertile categories were significantly associated with insulin sensitivity evaluated by the HOMA-R or the Matsuda index. Although it did not reach statistical significance, there was a trend toward increased pancreatic β-cell function evaluated by the HOMA-β with decreasing SgIo tertiles. The Steel–Dwass multiple comparisons tests revealed that plasma glucose and serum insulin levels at 2 h after a 75 g glucose load (OGTT PG 120 and OGTT IRI 120, respectively) in SgIo tertile 1 were significantly higher than in SgIo tertile 3 (*p* = 0.001 and 0.006, respectively). In addition, the Matsuda index of SgIo tertile 1 was lower than that of SgIo tertile 2 or of SgIo tertile 3 (*p* = 0.036 and 0.015, respectively).

### 3.2. The CGM Indices According to the SgIo Tertile Category

The effects of the SgIo tertile category on the CGM indices were examined next. In the participants in SgIo tertile 1, the median glucose variability evaluated by the standard deviation of CGM sensor glucose (CGM sd) was higher than those in SgIo tertile 2 or 3 (Table 2). Multiple comparisons with the Steel–Dwass test revealed that CGM sd was significantly higher in SgIo tertile 1 compared to tertile 2 (*p* = 0.046). CGM min and TBR70 were significantly associated with the SgIo tertile category (Table 2). Interestingly, the effects of the SgIo tertile category on the indices of low glucose levels, such as CGM min and TBR70, were biphasic. Post hoc analysis with the Steel–Dwass test revealed significant differences in TBR70 between SgIo tertiles 1 and 2 and SgIo tertiles 3 and 2. Compared to TBR70s in SgIo tertile 2, TBR70s in SgIo tertile 1 and 3 were significantly higher (*p* = 0.003 and 0.01, respectively).

### 3.3. The Proportions of Hypoglycemic Categories According to the SgIo Category

When the participants were divided either with or without CGM min < 70 mg/dL (3.9 mmol/L), TBR70 ≥ 1%, or SRH (TBR70 within 24 h after OGTT ≥ 0.6%), and the associations between these categories and the SgIo tertile category were examined, the proportions of the hypoglycemic categories in both SgIo tertiles 1 and 3 were higher than those in SgIo tertile 2, as shown in Table 3.

### 3.4. The Odds Ratios of SgIo Tertile 1 and 3 for Hypoglycemic Categories vs. SgIo Tertile 2

Including age, BMI, and fasting plasma glucose levels as covariates, multiple logistic regression models were next constructed in order to investigate whether there were any differences between SgIo tertiles 1 and 2, or between SgIo tertiles 3 and 2. Independently of the covariates, the SgIo category was significantly associated with hypoglycemic categories such as CGM min < 70 mg/dL (3.9 mmol/L), TBR ≥ 1%, or SRH in a biphasic manner. The participants in SgIo tertiles 1 and 3 had a greater risk of belonging to the hypoglycemic categories as compared to those in SgIo tertile 2. The odds ratios for belonging to the hypoglycemic categories in the participants in SgIo tertiles 1 and 3 compared to SgIo tertile 2 were approximately 7–12 (Table 4). When Fisher’s exact probability tests with post hoc Bonferroni corrections for multiple comparisons were applied, the chances of having SRH were significantly higher in the participants in SgIo tertiles 1 and 3 than in those in SgIo tertile 2 (*p* = 0.007 for SgIo tertile 1 vs. 2, and *p* = 0.007 for SgIo tertile 3 vs. 2).

### 3.5. The Association between SRH and SgIo Tertile Category According to the OGTT IRI 120 Tertile Category

The association between SRH and SgIo tertile category was next examined according to the tertile categories of post-challenge insulin levels. As shown in Table 5, in the lowest insulin category (OGTT IRI 120 tertile 1), there was a positive relationship between the proportion of the participants with SRH and the SgIo category. On the contrary, in the highest insulin group (OGTT IRI 120 tertile 3), there was a negative relationship between the proportion of the participants with SRH and the SgIo category. In the participants in OGTT IRI 120 tertile 1, the proportion of the participants with SRH was significantly higher in SgIo tertile 3 than in other SgIo tertiles (i.e., 1 and 2) (Fisher’s exact test *p* = 0.002). However, in the participants with OGTT IRI 120 tertile 3, the proportion of participants with SRH tended to be higher in SgIo tertile 1 than in other SgIo tertiles (i.e., 2 and 3) (Fisher’s exact test *p* = 0.028).

### 3.6. The Association between Snacking Habits and SgIo Tertile Categories

The association between self-reported snacking habits (a frequency of habitual snacking ≥ once a week) and SgIo category was next studied. In the present study, 36% of the study participants snacked regularly, at least once per week. As shown in Table 6, the proportion of participants with snacking habits was significantly associated with their SgIo category. The effects of the SgIo tertile category on the snacking habits category were biphasic, as were those observed for the hypoglycemic categories. When Fisher’s exact probability tests were applied, the participants in SgIo tertile 1 snacked significantly more frequently than those in SgIo tertile 2 (58.8% vs. 11.8%, *p* = 0.005), while the difference between SgIo tertiles 2 and 3 did not reach statistical significance (11.8% vs. 37.5%, *p* = 0.191). In addition, there were significant associations between the SgIo tertile category and the proportions of obesity (BMI ≥ 30), hyperinsulinemia (OGTT IRI 120 highest quartile), and impaired glucose tolerance (OGTT PG 120 ≥ 140 mg/dL (7.8 mmol/L)) (Table 6).

### 3.7. The Odds Ratios of SgIo Tertiles 1 and 3 vs. SgIo Tertile 2 for Having Snacking Habits, Obesity, Hyperinsulinemia, or Impaired Glucose Tolerance

The odds ratios for having snacking habits, obesity, hyperinsulinemia, or impaired glucose tolerance were then examined in the participants in SgIo tertile 1 and 3 compared to those in SgIo tertile 2. As shown in Table 7, the participants in SgIo tertile 1 were associated with 10.7-, 11.2-, 22.9-, and 13.8-fold higher risks of having snacking habits, obesity, hyperinsulinemia, and impaired glucose tolerance, respectively, compared to those in SgIo tertile 2. When Fisher’s exact probability tests were applied, the chances of having these conditions were significantly higher in the participants in SgIo tertile 1 than in those in SgIo tertile 2. Although it did not reach statistical significance, there was a trend toward higher odds for snacking habits in SgIo tertile 3 versus tertile 2. On the contrary, there was no significant difference between SgIo tertile 3 and SgIo tertile 2 with respect to the proportion of participants with BMI ≥ 30, OGTT IRI 120 highest quartile, or impaired glucose tolerance.

### 3.8. The Proportions of the Participants with Snacking Habits in the SgIo Tertile Categories According to the OGTT IRI 120 Tertile Category

As shown in Table 8, most (81.8%) of the participants in both SgIo tertile 1 and the highest OGTT IRI 120 tertile (OGTT IRI 120 tertile 3) categories had snacking habits. Among the participants in OGTT IRI 120 tertile 3, the proportion of participants with snacking habits was significantly higher in SgIo tertile 1 compared to those in the other SgIo categories (SgIo tertiles 2 and 3) (Fisher’s exact test, *p* = 0.018), with an odds ratio of 22.5 (95% CI; 1.6–314.6).

## 4. Discussion

In the present study, SgIo, the index of glucose effectiveness, was significantly associated with the hypoglycemic categories derived from CGM in a biphasic manner. Compared to the SgIo tertile 2 (middle) category, the tertile 1 (lowest) and tertile 3 (highest) SgIo categories had 7.3 times more chances of having CGM min < 70 mg/dL (3.9 mmol/L). When TBR70 within 24 h after OGTT ≥ 0.6% was defined as SRH, both SgIo tertiles 1 and 3 were associated with SRH (compared to the SgIo tertile 2), although the former, but not the latter, is associated with hyperinsulinemia. In the participants in the lowest SgIo tertile category, the proportions of snacking habits, obesity, and impaired glucose tolerance were higher than those of the participants in the other SgIo tertile categories.

Evaluation and management of hypoglycemia are recommended only in patients in whom Whipple’s triad—symptoms consistent with hypoglycemia, a low plasma glucose concentration, and resolution of the symptom(s) after the plasma glucose level is elevated—is documented [14]. In our recent study, half of the non-diabetic obese/overweight subjects exhibited minimal glucose levels of less than 70 mg/dL (3.9 mmol/L) within 24 h after OGTT, without notable hypoglycemic symptoms except hunger [5]. However, without Whipple’s triad (thus, subclinical), the glucose dip after a glycemic load was significantly associated with a higher eating/snacking frequency, suggesting that glucose levels that are in the hypoglycemic range, but above the threshold of neurogenic symptoms, induce appetite mostly without the patient’s awareness.

Previously, we have studied the role of SgIo in dysglycemia during CGM and found that lower glucose effectiveness is associated with the post-meal hyperglycemia observed in the daily life of obese/overweight men [12]. Since post-meal hyperglycemia in obesity without diabetes mellitus is often followed by an abrupt decrease in the glucose level, we examined the relationship between SgIo and SRH in the present study. In addition, it was investigated whether self-reported snacking habits were associated with low SgIo.

The present study revealed that there are two types of SRH, i.e., that with lower glucose effectiveness and that with higher glucose effectiveness. Since glucose effectiveness, per se, is the ability to increase peripheral glucose uptake and to decrease hepatic glucose production independently of insulin action [15], higher glucose effectiveness leads to higher glucose disposal, preventing a sudden increase in postprandial blood glucose. In fact, higher SgIo was associated with lower post-load blood glucose and insulin levels. After Roux-en-Y gastric bypass surgery for weight management, it is reported that, in addition to the increased β-cell response to oral stimuli, insulin-independent glucose disposal is also suggested to contribute to severe hypoglycemia [16]. This indicates that an increase in glucose effectiveness could play a crucial role in the establishment of symptomatic hypoglycemia, especially when gastric retention is reduced. Since SRH is significantly correlated with higher eating/snacking frequencies [5], a higher eating frequency in subjects with higher SgIo can be regarded as an innate protective mechanism to maintain normal glucose levels and to prevent symptomatic hypoglycemia.

On the other hand, subjects in the lower SgIo categories exhibited postprandial hyperglycemia [12]. As shown in Table 1 and Table 5, post-challenge hyperinsulinemia is associated with a lower SgIo category, suggesting that SRH in these subjects is dependent on insulin excess in response to hyperglycemia. Since lower SgIo was closely associated with higher BMI and insulin resistance (Table 1), the SRH-induced increase in appetite could lead to an excess of caloric intake, thus forming a vicious cycle leading to obesity. For obese/overweight subjects with low glucose effectiveness, SRH can be regarded as a link between obesity and appetite, and preventing SRH might be the key to controlling body weight.

The limitations of the study include the following. (1) Caution must be exercised to extrapolate the present finding to the general population because of the specific category (i.e., obese/overweight men) and the relatively small sample size which were evaluated. Particularly, when the participants were divided by both SgIo tertile category and OGTT IRI 120 tertile category, the numbers of some cells fell below one. Nonetheless, the analyses in this exploratory investigation were sufficient to obtain substantial ideas. (2) In the present study, we focused only on whether the participants had snacking habits, although the type, amount, frequency, and timing of snacking could also influence the glycemic response. (3) Since the present study only showed associations, it is not possible to determine causality or the direction of causality from the results themselves. In the literature, it is reported that mild hypoglycemia, the glucose levels of which are higher than the threshold of symptomatic autonomic activation [4] but sufficiently lower than that of appetite activation [17], could lead to an increased frequency of snacking, either consciously with hunger or subconsciously [5]. Ingesting sugary snacks could lead to postprandial hyperglycemia followed by hypoglycemia with an excess of insulin action, especially in people with lower glucose effectiveness [12]. Therefore, the causality of the association between snacking and SRH can be bi-directional, which suggests that the association between snacking and SRH creates a vicious cycle of obesity when glucose effectiveness is reduced. Well-controlled intervention studies are needed to demonstrate the hypotheses generated from the present study.

## 5. Conclusions

In obese or overweight individuals, SRH is associated with glucose effectiveness in a biphasic manner. Although the SRH-induced increase in appetite can be regarded as a protective mechanism to maintain normal glucose levels and prevent neuroglycopenia in subjects with higher glucose effectiveness, it may create a vicious cycle that leads to obesity in people with lower glucose effectiveness.

## Figures and Tables

**Table 1 metabolites-13-00238-t001:** Characteristics of the participants according to SgIo tertile category.

	SgIo Tertile 1		SgIo Tertile 2		SgIo Tertile 3		
	Median (IQR)	*n*	Median (IQR)	*n*	Median (IQR)	*n*	*p*
SgIo, mg/dL/min	2.02 (1.68–2.22)	17	2.59 (2.53–2.71)	17	2.96 (2.87–3.18)	16	
Age, years	56.0 (53.0–59.5)	17	54.0 (52.0–58.0)	17	53.5 (50.3–59.8)	16	0.705
BMI	29.0 (27.1–32.5)	17	27.8 (26.6–28.7)	17	26.8 (25.4–28.1)	16	0.009 *
HbA1c, %	5.6 (5.4–5.8)	17	5.3 (5.2–5.5)	17	5.3 (5.1–5.5)	16	0.017 *
1,5-AG, μg/mL	14.9 (10.7–23.7)	17	19.2 (15.4–23.8)	17	22.3 (19.4–26.6)	16	0.073
OGTT PG 0, mg/dL	92 (84.5–97.5)	17	90 (85.5–97.5)	17	91.5 (85.5–96.3)	16	0.972
OGTT PG 0, mmol/L	5.1 (4.7–5.4)		5.0 (4.8–5.4)		5.1 (4.8–5.4)		
OGTT PG 30, mg/dL	158 (138.8–178)	17	151 (137–177)	17	142 (119–171)	16	0.331
OGTT PG 30, mmol/L	8.8 (7.7–9.9)		8.4 (7.6–9.8)		7.9 (6.6–9.5)		
OGTT PG 60, mg/dL	170 (137–195)	17	177 (124–193)	17	148 (122–187)	16	0.434
OGTT PG 60, mmol/L	9.4 (7.6–10.8)		9.8 (6.9–10.7)		8.2 (6.8–10.4)		
OGTT PG 120, mg/dL	145 (119–163)	17	107 (96–129)	17	94.5 (77–111)	16	<0.001 *
OGTT PG 120, mmol/L	8.1 (6.6–9.1)		5.9 (5.3–7.2)		5.3 (4.3–6.2)		
OGTT IRI 0, μU/mL	10.1 (7.5–15.7)	17	7.2 (5.2–11.5)	17	6.2 (4.8–9.2)	16	0.026 *
OGTT IRI 30, μU/mL	70.3 (36.1–164)	17	58.2 (31.3–66.1)	17	49.3 (28.6–68.8)	16	0.265
OGTT IRI 60, μU/mL	65.9 (49.5–181)	17	60.8 (48.6–122)	17	68.2 (38.5–130)	16	0.629
OGTT IRI 120, μU/mL	131 (55.0–179)	17	48.9 (28.3–66.1)	17	37.1 (27.0–48.9)	16	0.002 *
HOMA-R	2.4 (1.9–3.6)	17	1.5 (1.1–2.5)	17	1.2 (1.0–2.2)	16	0.018 *
HOMA-β	130 (83.5–25)	17	101 (63.2–147)	17	89.6 (64.4–105)	16	0.052
Insulinogenic index	1.1 (0.5–2.3)	17	0.6 (0.4–1.0)	17	0.8 (0.5–1.4)	16	0.428
Matsuda index	3.3 (2.0–4.6)	17	4.7 (2.9–7.2)	17	5.4 (4.0–7.8)	16	0.008 *
Disposition index	3.1 (1.5–5.3)	17	3.7 (1.8–4.9)	17	3.8 (2.6–6.3)	16	0.373

SgIo, an OGTT-based index for glucose effectiveness; IQR, interquartile range; BMI, body mass index; 1,5-AG, 1,5-anhydroglucitol; OGTT, a 75 g oral glucose tolerance test; PG, plasma glucose concentration; IRI, immunoreactive serum insulin concentration; OGTT PG 0, 30, 60, and 120, PG at 0, 30, 60, and 120 min after a 75 g glucose load, respectively. OGTT IRI 0, 30, 60, and 120, IRI at 0, 30, 60, and 120 min after a 75 g glucose load, respectively. HOMA, homeostasis model assessment. Insulinogenic index, Matsuda index, and disposition index on postprandial PG were calculated as described in Methods. The Kruskal–Wallis tests were used to assess differences among the SgIo tertile categories. *: *p*-values < 0.05 were considered statistically significant.

**Table 2 metabolites-13-00238-t002:** The CGM indices according to the SgIo tertile category.

	SgIo Tertile 1		SgIo Tertile 2		SgIo Tertile 3		
	Median (IQR)	*n*	Median (IQR)	*n*	Median (IQR)	*n*	*p*
CGM mean	118 (111–123)	14	111 (104–120)	15	111 (105–118)	14	0.192
CGM max	209 (180–237)	14	187 (172–205)	15	177 (163–214)	14	0.153
CGM min	64.5 (47.8–70.8)	14	73 (67–81)	15	60 (50.8–69.3)	14	0.03 *
CGM sd	23.9 (21.7–28)	14	19 (16.7–21.5)	15	17.6 (15.7–21.8)	14	0.007 *
TBR70, %	1.91 (0.08–4.16)	14	0.0 (0.0–0.41)	15	1.04 (0.04–2.95)	14	0.01 *

SgIo, a 75 g oral glucose tolerance test (OGTT)-based index for glucose effectiveness; IQR, interquartile range; CGM, continuous glucose monitoring; CGM mean, max, min, and sd: the mean, maximum, minimum, and standard deviation of CGM sensor glucose levels during the study, respectively; TBR70, the percentage of time during which the CGM sensor glucose levels were below 70 mg/dL (3.9 mmol/L) during the study. The Kruskal–Wallis test was used to compare differences among the SgIo tertile categories. *: *p*-values < 0.05 were considered statistically significant.

**Table 3 metabolites-13-00238-t003:** The proportion of hypoglycemic categories according to the SgIo tertile category.

	SgIo Tertile 1		SgIo Tertile 2		SgIo Tertile 3		
	Percent	*n*/*n*	Percent	*n*/*n*	Percent	*n*/*n*	*p*
CGM min < 70 mg/dL	78.6%	11/14	33.3%	5/15	78.6%	11/14	0.014 *
TBR70 ≥ 1%	50%	7/14	13.3%	2/15	50%	7/14	0.06
SRH	64.3%	9/14	13.3%	2/15	64.3%	9/14	0.006 *

SgIo, an OGTT-based index for glucose effectiveness; CGM, continuous glucose monitoring; CGM min, minimal CGM sensor glucose levels during the study; TBR70, the percentage of time during which the CGM sensor glucose levels were below 70 mg/dL (3.9 mmol/L) during the study; SRH, subclinical reactive hypoglycemia, which was defined as TBR70 ≥ 0.6% within 24 h after a 75 g oral glucose tolerance test (OGTT). Pearson’s chi-squared test was used to determine whether there was an association between the proportions of participants belonging to hypoglycemic categories (i.e., CGM min < 70 mg/dL (3.9 mmol/L), TBR70 ≥ 1%, or SRH) and their SgIo tertile categories. *: *p*-values < 0.05 were considered statistically significant.

**Table 4 metabolites-13-00238-t004:** The odds ratios of SgIo tertiles 1 and 3 for belonging to the hypoglycemic categories vs. SgIo tertile 2.

	SgIo Tertile 1 vs. 2		SgIo Tertile 3 vs. 2	
	Odds (95% CI)	*p*	Odds (95% CI)	*p*
CGM min < 70 mg/dL	7.33 (1.38–38.9)	0.018 *	7.33 (1.38–38.9)	0.018 *
TBR70 ≥ 1%	6.5 (1.05–40.1)	0.041	6.5 (1.05–40.1)	0.041
SRH	11.7 (1.85–74.2)	0.007 *	11.7 (1.85–74.2)	0.007 *

SgIo, an OGTT-based index for glucose effectiveness; CI, confidence interval; CGM, continuous glucose monitoring; CGM min, minimal CGM sensor glucose levels; TBR70, the percentage of time during which the CGM sensor glucose levels were below 70 mg/dL (3.9 mmol/L) during the study; SRH, subclinical reactive hypoglycemia, which was defined as TBR70 ≥ 0.6% within 24 h after a 75 g oral glucose tolerance test (OGTT). Fisher’s exact probability tests with post hoc Bonferroni corrections for multiple comparisons were applied to investigate whether there were any differences between SgIo tertiles 1 and 2, or between SgIo tertiles 3 and 2. *: *p*-values < 0.025 were considered statistically significant.

**Table 5 metabolites-13-00238-t005:** The proportions of the participants with SRH in the SgIo tertile categories according to OGTT IRI 120 tertile category.

	SgIo Tertile 1		SgIo Tertile 2		SgIo Tertile 3		
	Percent	*n*/*n*	Percent	*n*/*n*	Percent	*n*/*n*	*p* For Trend
OGTT IRI 120 tertile 1	0%	0/1	0%	0/5	87.5%	7/8	0.001 *
OGTT IRI 120 tertile 2	75.0%	3/4	28.6%	2/7	50%	2/4	0.239
OGTT IRI 120 tertile 3	66.7%	6/9	0%	0/3	0%	0/2	0.013 *

SgIo, an OGTT-based index for glucose effectiveness; OGTT, a 75 g oral glucose tolerance test; IRI, immunoreactive serum insulin concentration; OGTT IRI 120, IRI at 120 min after a 75 g glucose load. The proportions of the participants with SRH in the SgIo tertile categories according to the OGTT IRI 120 tertile category are shown. SRH, subclinical reactive hypoglycemia, which was defined as TBR70 ≥ 0.6% within 24 h after OGTT. Cochran–Armitage trend *p* was calculated for each OGTT IRI 120 tertile. *: *p*-values < 0.05 were considered statistically significant.

**Table 6 metabolites-13-00238-t006:** The proportions of the participants with snacking habits, obesity, hyperinsulinemia, and dysglycemia, according to SgIo tertile category.

	SgIo Tertile 1		SgIo Tertile 2		SgIo Tertile 3		
	Percent	*n*/*n*	Percent	*n*/*n*	Percent	*n*/*n*	*p* for Trend
Snacking habits	58.8%	10/17	11.8%	2/17	37.5%	6/16	0.017 *
BMI ≥ 30	41.2%	7/17	5.9%	1/17	6.3%	1/16	0.009 *
OGTT IRI 120 highest quartile	58.8%	10/17	5.9%	1/17	6.3%	1/16	<0.001 *
Impaired glucose tolerance	64.7%	11/17	11.8%	2/17	0%	0/16	<0.001 *

SgIo, an OGTT-based index for glucose effectiveness; BMI, body mass index; OGTT, a 75 g oral glucose tolerance test; IRI, immunoreactive serum insulin concentration; OGTT IRI 120, IRI at 2 h after a 75 g glucose challenge; impaired glucose tolerance is defined by plasma glucose levels at 2 h after a 75 g glucose challenge ≥ 140 mg/dL (7.8 mmol/L). The Pearson’s chi-squared test was used to determine whether there was a significant difference among the SgIo tertile categories with respect to the proportions of snacking habits, obesity, hyperinsulinemia, and dysglycemia. *: *p*-values of less than 0.05 were considered statistically significant.

**Table 7 metabolites-13-00238-t007:** The odds ratios of the SgIo tertile categories (vs. SgIo tertile 2) for having snacking habits, obesity, hyperinsulinemia, or impaired glucose tolerance.

	SgIo Tertile 1 vs. 2		SgIo Tertile 3 vs. 2	
	Odds (95% CI)	*p*	Odds (95% CI)	*p*
Snacking habits	10.7 (1.84–62.5)	0.005 *	4.5 (0.75–26.9)	0.093
BMI ≥ 30	11.2 (1.19–105.1)	0.02 *	1.07 (0.06–18.6)	0.742
OGTT IRI 120 highest quartile	22.9 (2.44–214.6)	0.001 *	1.07 (0.06–18.6)	0.742
Impaired glucose tolerance	13.8 (2.32–81.5)	0.002 *	-	-

SgIo, an OGTT-based index for glucose effectiveness; CI, confidence interval; BMI, body mass index; OGTT, a 75 g oral glucose tolerance test; IRI, immunoreactive serum insulin concentration; OGTT IRI 120, IRI at 2 h after a 75 g glucose challenge; impaired glucose tolerance is defined by plasma glucose levels at 2 h after a 75 g glucose challenge ≥ 140 mg/dL (7.8 mmol/L). Fisher’s exact probability tests with post hoc Bonferroni corrections for multiple comparisons were applied to investigate whether there were any differences between SgIo tertiles 1 and 2, and between SgIo tertiles 3 and 2. *: For multiple comparisons, *p*-values of less than 0.025 were considered statistically significant.

**Table 8 metabolites-13-00238-t008:** The proportions of the participants with snacking habits in the SgIo tertile categories according to OGTT IRI 120 tertile category.

	SgIo Tertile 1		SgIo Tertile 2		SgIo Tertile 3		
	Percent	*n*/*n*	Percent	*n*/*n*	Percent	*n*/*n*	*p* vs. for Trend
OGTT IRI 120 tertile 1	0%	0/2	16.7%	1/6	37.5%	3/8	0.107
OGTT IRI 120 tertile 2	25%	1/4	14.3%	1/7	33.3%	2/6	0.345
OGTT IRI 120 tertile 3	81.8%	9/11	0	0/4	50%	1/2	0.028 *

SgIo, an OGTT-based index for glucose effectiveness; OGTT, a 75 g oral glucose tolerance test; IRI, immunoreactive serum insulin concentration; OGTT IRI 120, IRI at 120 min after a 75 g glucose load. The proportions of participants with snacking habits in the SgIo tertile categories according to OGTT IRI 120 tertile category are shown. Cochran–Armitage trend *p* was calculated in each OGTT IRI 120 tertile. *: *p*-values < 0.05 were considered statistically significant.

## Data Availability

The data are not publicly available due to privacy reasons and ethical restrictions.

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
