# Peer review of "Lower Glucose Effectiveness Is Associated with Subclinical Reactive Hypoglycemia, Snacking Habits, and Obesity"

_metabolites, 2023, doi:10.3390/metabo13020238_

Round 1

Reviewer 1 Report

Abstract

-It would be helpful if authors would include a concluding statement that eludes to how these data could be used to help improve biomedicine

Introduction

-Broad grammar check is needed

-Similar to the comment on the abstract, a statement at the end of the introduction is needed to explain to the reader why this study is important.

Methods

-Was exercise/physical activity controlled for?

-Why were mid-life adults recruited for this study? This should be justified in the manuscript

Results

-Did obesity status/BMI alone predict glucose effectiveness?

-Any information about what individuals snacked on? There are lines of evidence to suggest that the type of snack influences the glycemic response (PMID: 30952505 among others)

Author Response

To reviewer 1

Thank you for your thoughtful comments. According to your suggestions, we extensively revised our manuscript as shown below in a point-by-point manner.

Q1. It would be helpful if authors would include a concluding statement that eludes to how these data could be used to help improve biomedicine

A1. According to your comments, we added the following sentence in the revised manuscript.

In the Abstract,

“In conclusion, SgIo is associated with SRH in a biphasic manner. In people with lower glucose effectiveness, the SRH-induced increase in appetite may create a vicious cycle that leads to obesity.”

Q2. Broad grammar check is needed.

A2. To increase clarity, we extensively rewrote the Introduction section. In addition, we carefully performed a grammar check with Grammarly. We also checked if similar expressions can be found in the literature.

Q3. Similar to the comment on the abstract, a statement at the end of the introduction is needed to explain to the reader why this study is important.

A3. We added the following sentence in the revised manuscript.

In the Introduction,

“However, the role of glucose effectiveness on SRH has not been studied. In the present study, we, therefore, investigated the relationship between SgIo and SRH.”

In addition, we rewrote the Introduction section, in order to explain the significance of the study more clearly.

“…. Therefore, SRH could play a key role in the establishment and/or maintenance of obesity, and studying the mechanism of SRH is of clinical significance.”

Q4. Was exercise/physical activity controlled for?

A4. Exercise/physical activity was not controlled. These were at their own discretion.

In the revised manuscript, we added the following sentence at the end of the first paragraph of the Methods section.

“During the study, eating, drinking, and exercise were at the discretion of the participants.”

Q5. Why were mid-life adults recruited for this study? This should be justified in the manuscript

A5.We recruited mid-life men with obesity or overweight since the population is at high risk for developing future diabetes.

In methods,

“… were recruited (mid-life men with obesity or overweight were studied since the population is at high risk for developing future diabetes).”

Q6.  Did obesity status/BMI alone predict glucose effectiveness?

A6. As shown in Table 6, the proportion of obesity is significantly higher in the lowest SgIo tertile. In addition, there is a significant linear association between lower SgIo and higher BMI. The present study proposes that asymptomatic reactive hypoglycemia might be a link between lower SgIo and obesity/overweight.

Q7. Any information about what individuals snacked on? There are lines of evidence to suggest that the type of snack influences the glycemic response (PMID: 30952505 among others)

A7. As the reviewer pointed out, the type of snack definitely influences the glycemic response. We have all data regarding what individuals snacked on during the study. Although we focused on only the frequency of snacking in the present and previous studies, we would like to study the contents of snacking in the next study. We added the important point as a limitation in the Discussion section.

“…2) In the present study, we focused on if the participants have snacking habits, while the types, amount, frequency, or timing of snacking could also influence the glycemic response, and 3)…”

Reviewer 2 Report

Thank you for submitting an interesting evaluation of CGMS profiles in relation to glucose effectiveness. I have some comments / questions to be addressed below:

-The standard unit for glucose concentration is mmol/l rather than mg/dl.

-please use the term 'participants' rather than 'subjects' throughout the manuscript.

-line 32 'diabetes' should be clarified as diabetes mellitus

-line 32 when referencing 'hypoglycaemia' is this referring to symptoms of hypoglycaemia or blood/plasma/interstitial glucose concentration in the hypoglycaemic range? Please clarify in the text & check throughout that this is clearly described.

-Reference 2: it is not appropriate to be referencing a literature review/bulletin; the earlier reports/source data should be identified so that those who have carried out the investigations are credited for their work. The exception to this would only be if a review is providing additional insight and it is this insight which you are specifically referencing. Please check all references used to ensure this is followed.

-line 41: please clarify that this information was determined during a hyperinsulinaemic clamp i.e. "Using the hyperinsulinaemic clamp, activation of autonomic symptoms etc..."

-line 48/49 how was direction of causality determined from associations? Snacking habits may arise from behavioural drivers independent of blood glucose concentration / dynamics and snacking on high sucrose/glucose foods may lead to greater excursions in blood glucose concentration i.e. direction of causality may not be in the direction you state. Please be specific throughout the manuscript about what can and what cannot be inferred from data associations. 

-line 52; it is also likely that those reporting symptoms of hypoglycaemia are a heterogenous cohort with different underlying triggers for their reactive hypoglycaemia.

-the use of the word 'levels' throughout the manuscript is not appropriate. Please be specific about what you are meaning i.e. concentration. Also, where glucose concentration is referred to, please specify whether this is blood/plasma/interstitial 

- line 60: please define SgIo and its units

-sentence 62-64: this is convoluted and doesn't make sense - suspect that there are typographical errors present.

-please define the acronym PPG the first time it is used.

- for the insulin related indices (method section), please indicate the units that insulin concentration is being measured in.

-throughout the result section, please replace the use of the word 'insignificant' with 'not reaching statistical significance' to make it clear as to what you want to convey.

- the words 'tendency' usually refers to a P value above 0.05, but below 0.1. The comparisons identified as tending towards something were all P>0.1.

 - please present the SgIo ranges for each tertile in the result section.

- it is not clear what you mean by 'lower'/ 'higher' SgIo group as the SgIo ranges have not been presented. Does 'low' mean tertile number i.e. tertile 1? Please be specific as to which tertile you are referring to throughout ie. tertile 1, 2 or 3 rather than 'lower' 'higher'. 

-Please describe the snacking habits of the cohort in greater detail. What were the items in the FFQ.  What were the grouping characteristics e.g. high sugar, high fat etc. Can this be included as supplementary material? Were snack types related to measures or just eating frequency?

- Did you just collect frequency of eating/ snacking or also the timing of eating / snack consumption in relation to the cgms recording?

-line 322 ('independent'/dependent on') suggests causation. only associations are presented so it is not possible to infer causation. please be more specific and only report within the limits of the data. In the same line direction of causality can not be determined from associations therefore it cannot be stated that low interstitial glucose concentration caused greater snacking unless snacking/eating was reported by participants at the time that concentration was in the hypo range. 

- the limitations section should be expanded to encompass some of the above points i.e. that it is not possible to determine causality or direction of causality from associations and that well controlled intervention studies are needed to investigate your hypotheses. Also that timing of eating has not been measured in relation to GCMS data (if this is the case) to be able to speculate on the relationship between the 2 parameters.

Author Response

To reviewer 2

Thank you for your helpful comments and suggestive questions. We extensively revised our manuscript as shown below in a point-by-point manner.

Q1.The standard unit for glucose concentration is mmol/l rather than mg/dl.

A1. Since both units are used in the global literature, we added mmol/l in addition to mg/dl in all data.

Q2. Please use the term 'participants' rather than 'subjects' throughout the manuscript.

A2. We changed the term from 'subjects' to 'participants', when ‘subjects’ means participants in the present study.

Q3. line 32 'diabetes' should be clarified as diabetes mellitus. line 32 when referencing 'hypoglycaemia' is this referring to symptoms of hypoglycaemia or blood/plasma/interstitial glucose concentration in the hypoglycaemic range? Please clarify in the text & check throughout that this is clearly described.

A3. We changed from ‘diabetes’ to ‘diabetes mellitus’ throughout the manuscript except “type 2 diabetes”. In general, ‘hypoglycemia’ is defined as low glucose concentrations with the Whipple’s triad. In the revised manuscript, we increased the clarity of the important point raised by the reviewer by differentiating if the 'hypoglycemia' is referring to that with or without typical symptoms. We added ‘symptomatic’ to the most of the 'hypoglycemia' if it means clinical hypoglycemia, while the ‘glucose concentrations in the hypoglycemic range without typical hypoglycemic symptoms after a 75g-glucose load’ are termed as SRH, subclinical reactive hypoglycemia.

Q4. Reference 2: it is not appropriate to be referencing a literature review/bulletin; the earlier reports/source data should be identified so that those who have carried out the investigations are credited for their work. The exception to this would only be if a review is providing additional insight and it is this insight which you are specifically referencing. Please check all references used to ensure this is followed.

A4. We changed Reference 2 to the original paper shown below.

“Brun, JF.; Fedou, C.; Mercier, J.; Postprandial reactive hypoglycemia. Diabetes Metab. 2000, 26, 337–51.”

Q5. line 41: please clarify that this information was determined during a hyperinsulinaemic clamp i.e. "Using the hyperinsulinaemic clamp, activation of autonomic symptoms etc..."

A5. We added the method as you suggested.

In the second paragraph of the Introduction section,

“Using the hyperinsulinemic glucose clamp, activation of autonomic symptoms such as palpitation, tremor, and sweating develop at a plasma glucose concentration of approximately 58 mg/dL (3.2 mmol/L) [4].”

Q6. line 48/49 how was direction of causality determined from associations? Snacking habits may arise from behavioural drivers independent of blood glucose concentration / dynamics and snacking on high sucrose/glucose foods may lead to greater excursions in blood glucose concentration i.e. direction of causality may not be in the direction you state. Please be specific throughout the manuscript about what can and what cannot be inferred from data associations. 

A6. The previous studies suggest that the glucose threshold for typical symptoms of hypoglycemia is lower than that for activation of appetite. For example, with the use of the hyperinsulinemic glucose clamp, it is reported that activation of the autonomic symptoms such as anxiety, palpitations, sweating, irritability, and tremor began at plasma glucose concentrations of 58 mg/dL [Am. J. Physiol. Metab. 1991, 260, E67–E74], while functional MRI combined with a stepped hyperinsulinemic euglycemic–hypoglycemic clamp revealed that mild hypoglycemia of 67 +/− 1 mg/dL preferentially activates limbic-striatal brain regions in response to food cues to produce a greater desire for high-calorie foods [J. Clin. Invest. 2011, 121, 4161–4169]. It is, therefore, possible that mild hypoglycemia, the glucose levels of which are higher than the threshold of symptomatic autonomic activation but sufficiently lower than that for appetite activation, might lead to an increased frequency of snacking, either consciously or subconsciously, to prevent further hypoglycemia. As you commented, snacking could very well lead to glucose excursions. Therefore, the causality for the association between snacking and SRH can be bi-directional. In fact, this is the point of our previous paper [Endocrines 2022, 3, 530–537], which suggests the association between snacking and SRH makes a vicious cycle of obesity. We added the above points as the limitation of the present study in the Discussion section.

In the last paragraph of the Discussion section,

“… and 3) Since the present study only shows the association, it is not possible to determine causality or direction of causality from the results themselves. In the literature, it is reported that mild hypoglycemia, the glucose levels of which are higher than the threshold of symptomatic autonomic activation [4] but sufficiently lower than that for appetite activation [17], could lead to an increased frequency of snacking, either consciously with hunger or subconsciously [5], while ingestion of sugary snacks could lead to hypoglycemia following to postprandial hyperglycemia, especially in people with lower glucose effectiveness [12]. Therefore, the causality for the association between snacking and SRH can be bi-directional, which suggests the association between snacking and SRH makes a vicious cycle of obesity when glucose effectiveness is reduced....”

Q7. line 52; it is also likely that those reporting symptoms of hypoglycaemia are a heterogenous cohort with different underlying triggers for their reactive hypoglycaemia.

A7. As you suggested, the specific causes have not been fully determined and the underlying triggers of hypoglycemia could be heterogeneous. Actually, this is the reason why we started the current analysis. Although too much insulin in the bloodstream at the incorrect timing is thought to be one of the major causes of symptomatic reactive hypoglycemia, the present paper proposes that dysregulated glucose effectiveness could be another underlying cause.

Q8. the use of the word 'levels' throughout the manuscript is not appropriate. Please be specific about what you are meaning i.e. concentration. Also, where glucose concentration is referred to, please specify whether this is blood/plasma/interstitial 

A8. As you commented, the CGM sensor is designed to collect the glucose information in the fluid just under the skin (interstitial fluid) and send a reading, which is retrieved by the transmitter. We added the information in the Methods section of the revised manuscript. We prefer using the word “levels” since many other studies utilizing CGM as well as the maker (Medtronic) have been using the word “glucose levels”.

In the first paragraph of the Methods section,

“… The CGM sensor is designed to collect the glucose information in the interstitial fluid and send a reading, which is retrieved by the transmitter. The sensor was ..”

Q9. line 60: please define SgIo and its units

A9. SgIo is not an abbreviation. ‘S’, ‘g’, ‘I’, and ‘o’ stand for ‘sensitivity’, ‘glucose’, ’index’, and ’oral’, respectively.  “A 75-g oral glucose tolerance test (OGTT)-derived index for glucose effectiveness” is the definition of SgIo. As you instructed, we added the unit of SgIo, mg/dl/min, in the Methods, Results, and Table 1 of the revised manuscript.

Q10. sentence 62-64: this is convoluted and doesn't make sense - suspect that there are typographical errors present.

A10. To reduce the risk of misunderstanding, we changed the end of the Introduction more simply to “However, the role of glucose effectiveness on SRH has not been studied. In the present study, we, therefore, investigated the relationship between SgIo and SRH.”

Q11. please define the acronym PPG the first time it is used.

A11. We define PPG as “post-loading plasma glucose” for the first time used in the revised manuscript.

Q12.  for the insulin related indices (method section), please indicate the units that insulin concentration is being measured in.

A12. We indicate the unit of insulin in the revised manuscript.

Q13. throughout the result section, please replace the use of the word 'insignificant' with 'not reaching statistical significance' to make it clear as to what you want to convey.

A13. We changed the phrase as you suggested.

Q14. the words 'tendency' usually refers to a P value above 0.05, but below 0.1. The comparisons identified as tending towards something were all P>0.1.

A14. We removed the word ‘tendency’ from the revised manuscript.

Q15. please present the SgIo ranges for each tertile in the result section.

A15. We added the ranges of SgIo tertile categories in the Result section. In addition, the medians and IQRs of each tertile were presented in Table 1.

Q16. it is not clear what you mean by 'lower'/ 'higher' SgIo group as the SgIo ranges have not been presented. Does 'low' mean tertile number i.e. tertile 1? Please be specific as to which tertile you are referring to throughout ie. tertile 1, 2 or 3 rather than 'lower' 'higher'. 

A16. In the revised manuscript, we changed from 'lower', ‘middle’ or 'higher' tertile to tertile 1, 2, or 3.

Q17. Please describe the snacking habits of the cohort in greater detail. What were the items in the FFQ.  What were the grouping characteristics e.g. high sugar, high fat etc. Can this be included as supplementary material? Were snack types related to measures or just eating frequency? - Did you just collect frequency of eating/ snacking or also the timing of eating / snack consumption in relation to the cgms recording?

A17. As the reviewer pointed out, the types, amounts, frequency, and timing of snacking are important. In the present study, we focused only on self-reported snacking habits. We included the point in the limitation.

In the Discussion section,

“.. 2) In the present study, we focused on if the participants have snacking habits, while the types, amount, frequency, or timing of snacking could also influence the glycemic response, and 3) …”

Q18. line 322 ('independent'/dependent on') suggests causation. only associations are presented so it is not possible to infer causation. please be more specific and only report within the limits of the data.

A18. In the revised manuscript, we changed the phrase to “.. both the SgIo tertile 1 and 3 are associated with SRH (compared to the SgIo 2), while not the former but the latter is associated with hyperinsulinemia.”

Q19. In the same line direction of causality can not be determined from associations therefore it cannot be stated that low interstitial glucose concentration caused greater snacking unless snacking/eating was reported by participants at the time that concentration was in the hypo range. 

A19. Same as in A6. In fact, snacking/eating was documented by participants not only at the time that concentration was in the hypo range but also during at least a few days after the episode of hypoglycemia [Endocrines 2022, 3(3), 530-537]. Therefore, it seems that the memory of hypoglycemia will last for a certain period.

Q20. the limitations section should be expanded to encompass some of the above points i.e. that it is not possible to determine causality or direction of causality from associations and that well controlled intervention studies are needed to investigate your hypotheses. Also that timing of eating has not been measured in relation to GCMS data (if this is the case) to be able to speculate on the relationship between the 2 parameters.

A20. Thank you for the insightful comments and your points are well taken. In the revised manuscript, we expanded the limitation section in the last paragraph of the Discussion as below.

“The limitation of the study includes 1) Caution must be exercised to extrapolate the present finding to the general population because of the specific category (i.e., obese/overweight men) and the relatively small sample size. Particularly, when participants were divided with both the SgIo tertile category and OGTT IRI 120 tertile category, the numbers of some cells fell below one. Nonetheless, the analyses seem sufficient for this exploratory investigation to obtain substantial ideas, 2) In the present study, we focused on if the participants have snacking habits, while the types, amount, frequency, or timing of snacking were not taken into account, and 3) Since the present study only shows the association, it is not possible to determine causality or direction of causality from the results themselves. In the literature, it is reported that mild hypoglycemia, the glucose levels of which are higher than the threshold of symptomatic autonomic activation [4] but sufficiently lower than that for appetite activation [17], could lead to an increased frequency of snacking, either consciously with hunger or subconsciously [5], while ingestion of sugary snacks could lead to hypoglycemia following to postprandial hyperglycemia, especially in people with lower glucose effectiveness [12]. Therefore, the causality for the association between snacking and SRH can be bi-directional, which suggests the association between snacking and SRH makes a vicious cycle of obesity when glucose effectiveness is reduced. Well-controlled intervention studies are needed to demonstrate the hypotheses generated from the present study.”
